# Variant Analysis of SARS-CoV-2 Genomes from Belgian Military Personnel Engaged in Overseas Missions and Operations

**DOI:** 10.3390/v13071359

**Published:** 2021-07-13

**Authors:** Jean-Paul Pirnay, Philippe Selhorst, Samuel L. Hong, Christel Cochez, Barney Potter, Piet Maes, Mauro Petrillo, Gytis Dudas, Vincent Claes, Yolien Van der Beken, Gilbert Verbeken, Julie Degueldre, Simon Dellicour, Lize Cuypers, France T’Sas, Guy Van den Eede, Bruno Verhasselt, Wouter Weuts, Cedric Smets, Jan Mertens, Philippe Geeraerts, Kevin K. Ariën, Emmanuel André, Pierre Neirinckx, Patrick Soentjens, Guy Baele

**Affiliations:** 1Laboratory for Molecular and Cellular Technology, Queen Astrid Military Hospital, 1120 Brussels, Belgium; christel.cochez@mil.be (C.C.); Gilbert.verbeken@mil.be (G.V.); 2Unit of Virology and Outbreak Research Team, Department of Biomedical Sciences, Institute of Tropical Medicine, 2000 Antwerp, Belgium; pselhorst@itg.be; 3Department of Microbiology, Immunology and Transplantation, Rega Institute, KU Leuven, 3000 Leuven, Belgium; samuel.hong@kuleuven.be (S.L.H.); barneyisaksen.potter@kuleuven.be (B.P.); piet.maes@kuleuven.be (P.M.); simon.dellicour@ulb.ac.be (S.D.); guy.baele@kuleuven.be (G.B.); 4European Commission, Directorate-General Joint Research Centre (JRC), 21027 Ispra, Italy; mauro.petrillo@ec.europa.eu; 5Gothenburg Global Biodiversity Centre, 413 19 Gothenburg, Sweden; gytisdudas@gmail.com; 6Hematology, Oncology and Transfusion Medicine Center, Vilnius University Hospital Santaros Klinikos, 08410 Vilnius, Lithuania; 7Clinical Laboratory, Queen Astrid Military Hospital, 1120 Brussels, Belgium; vincent.claes@mil.be (V.C.); yolien.vanderbeken@mil.be (Y.V.d.B.); julie.degueldre@mil.be (J.D.); france.tsas@mil.be (F.T.); 8Spatial Epidemiology Lab (SpELL), Université Libre de Bruxelles, 1050 Bruxelles, Belgium; 9Department of Laboratory Medicine, UZ Leuven Hospital, 3000 Leuven, Belgium; lize.cuypers@uzleuven.be (L.C.); emmanuel.andre@kuleuven.be (E.A.); 10European Commission, Directorate-General Joint Research Centre (JRC), 1050 Brussels, Belgium; guy.van-den-eede@ec.europa.eu; 11Department of Diagnostic Sciences, Ghent University Hospital, Ghent University, 9000 Ghent, Belgium; bruno.verhasselt@ugent.be; 12Queen Astrid Military Hospital, 1120 Brussels, Belgium; wouter.weuts@mil.be; 1314th Medical Battalion, 1800 Peutie, Belgium; cedric.smets@mil.be; 14Medical Component, Ministry of Defense, 1140 Brussels, Belgium; jan.mertens@mil.be (J.M.); philippe.geeraerts@mil.be (P.G.); pierre.neirinckx@mil.be (P.N.); 15Unit of Virology, Department of Biomedical Sciences, Institute of Tropical Medicine, 2000 Antwerp, Belgium; karien@itg.be; 16Department of Biomedical Sciences, University of Antwerp, 2610 Antwerp, Belgium; 17Laboratory of Clinical Bacteriology and Mycology, Department of Microbiology, Immunology and Transplantation, Rega Institute, KU Leuven, 3000 Leuven, Belgium; 18Center for Infectious Diseases, Queen Astrid Military Hospital, 1120 Brussels, Belgium; patrick.soentjens@mil.be; 19Department of Clinical Sciences, Institute of Tropical Medicine, 2000 Antwerp, Belgium

**Keywords:** SARS-CoV-2, COVID-19, military, outbreak, variants, genomic epidemiology

## Abstract

More than a year after the first identification of severe acute respiratory syndrome coronavirus 2 (SARS-CoV-2) as the causative agent of the 2019 coronavirus disease (COVID-19) in China, the emergence and spread of genomic variants of this virus through travel raise concerns regarding the introduction of lineages in previously unaffected regions, requiring adequate containment strategies. Concomitantly, such introductions fuel worries about a possible increase in transmissibility and disease severity, as well as a possible decrease in vaccine efficacy. Military personnel are frequently deployed on missions around the world. As part of a COVID-19 risk mitigation strategy, Belgian Armed Forces that engaged in missions and operations abroad were screened (7683 RT-qPCR tests), pre- and post-mission, for the presence of SARS-CoV-2, including the identification of viral lineages. Nine distinct viral genotypes were identified in soldiers returning from operations in Niger, the Democratic Republic of the Congo, Afghanistan, and Mali. The SARS-CoV-2 variants belonged to major clades 19B, 20A, and 20B (Nextstrain nomenclature), and included “variant of interest” B.1.525, “variant under monitoring” A.27, as well as lineages B.1.214, B.1, B.1.1.254, and A (pangolin nomenclature), some of which are internationally monitored due to the specific mutations they harbor. Through contact tracing and phylogenetic analysis, we show that isolation and testing policies implemented by the Belgian military command appear to have been successful in containing the influx and transmission of these distinct SARS-CoV-2 variants into military and civilian populations.

## 1. Introduction

The 2019 coronavirus disease (COVID-19) is an infectious disease caused by severe acute respiratory syndrome coronavirus 2 (SARS-CoV-2), a single-stranded RNA beta-coronavirus with a genome 29,903 nucleotides long [1]. First identified at the end of 2019 in Wuhan, China, it spread rapidly and caused the ongoing global pandemic. Even though the disease predominantly causes lung damage, other organs such as the heart can also be affected [2]. At the time of writing (17 April 2021), 175,987,176 confirmed COVID-19 cases, including 3,811,561 deaths, had been reported to the World Health Organization (WHO) worldwide [3]. Although coronaviruses have a genetic proofreading mechanism, viral mutations occur and can spread and increase in frequency, mainly due to natural selection of beneficial mutations. SARS-CoV-2 protein sequence diversity may be associated with pathogenicity and/or transmission of the virus, and certain SARS-CoV-2 genotypes or variants were found to be specific to geographical locations [4]. Therefore, it has been suggested that genomic, epidemiological, and clinical data should be combined to better understand the infectivity and pathogenicity of SARS-CoV-2 [5].

Today, the risk associated with the international spread of new SARS-CoV-2 variants (via travel) is a matter of great concern. In November 2020, a new SARS-CoV-2 variant, named B.1.1.7 or 20I/501Y.V1 (depending on the nomenclature used), was identified in the United Kingdom (UK) [6,7]. In December 2020, it was reported that the transmission rate of this variant could be up to 71% higher than for other variants [7]. This new variant was subsequently identified in several other countries and has become globally dominant today. It was recently suggested that the B.1.1.7 variant causes higher COVID-19 mortality [8], and it was found to be associated with higher viral load and younger age of infected patients [9]. Another highly transmissible SARS-CoV-2 variant named B.1.351 (or 20H/501Y.V2) likely emerged from the first wave of the South African COVID-19 epidemic and then spread quickly to its neighboring country of Botswana in December 2020, as well as several other countries across the world in January 2021 [10]. There are concerns that certain mutations in the spike protein (e.g., K417N, E484K, N501Y) could decrease the effectiveness of COVID-19 vaccines. An increased resistance of variant B.1.351, but also B.1.1.7, to antibody neutralization has been reported [11]. More recently, Wang et al. reported that an emergent variant from Brazil, named P.1 or 20J/501Y.V3, was also more resistant to neutralization by convalescent plasma and vaccine sera [12]. Even though a small study examining the impact of the N501Y mutation on the Pfizer-BioNTech vaccine did not show any loss of antibody neutralization efficacy, further studies examining the impact of other mutations (e.g., K417N, E484K, …), and involving the other COVID-19 vaccines, are required [13].

The COVID-19 pandemic also poses a threat to the readiness of the Armed Forces and their ability to conduct military operations [14], especially in close contact environments in military bases or on warships [15]. In addition, they are frequently involved in operations and overseas missions around the world, resulting in global movements of military personnel. The Belgian military command has implemented organizational changes and preventive measures to prevent the introduction and spread of SARS-CoV-2 among troops, especially during missions and operations abroad, and among their colleagues, their family, and the civilian population upon their return to Belgium. 

In this study, we focused on Belgian military personnel involved in overseas missions and operations during the period from May 2020 to April 2021. A flowchart detailing operational procedures with respect to the imposed corona measures can be seen in Figure 1. Military service members were placed in a controlled quarantine facility, in small cohorts in separate parts of the facility, for a period of 14 days before their deployment abroad. They had no contact with anyone outside their cohort, with the exception of dedicated military medical personnel (e.g., to monitor for COVID-19). Two days before their scheduled departure, nasopharyngeal swabs were collected for determination of possible SARS-CoV-2 viral load using quantitative RT-PCR (RT-qPCR). One day before deployment, each soldier underwent a medical exam and only those with a negative test result and showing no COVID-19 symptoms were admitted to the mission. Since 1 May 2021, efforts are being made to staff overseas military theatres only with soldiers who have been fully vaccinated for COVID-19, allowing this quarantine measure to be lifted (Figure 1), unless the destination country or the mission’s lead nation request otherwise. During missions, social distancing and source control (e.g., wearing surgical masks) have been sought, but sometimes this was difficult to reconcile with daily tasks in confined spaces and the need to train or fight. Regular contacts with local and international colleagues were also required. Relevant Belgian military operational settings, such as training camps in austere environments and warships, were equipped with mobile infectious diseases (incl. COVID-19) testing capabilities (GeneXpert systems) for rapid detection of SARS-CoV-2.

Soldiers who test positive during a mission abroad and are not vaccinated for COVID-19 are automatically repatriated to Belgium within seven days; vaccinated soldiers upon medical decision only (Figure 1). Upon their return to Belgium, all soldiers complied with the national testing, isolation, and contact tracing measures. During the study period, this meant that returning soldiers were sampled and tested for SARS-CoV-2 viral load at days 0 and 7 and were required to quarantine themselves at home for 10 days. If the test was positive on day 7, self-isolation was required for at least 7 more days. Persons considered as high-risk contacts of a SARS-CoV-2-positive soldier were subjected to the same regimen. In response to the increased incidence of SARS-CoV-2 variants of concern (VOCs), in December 2020, the post-mission quarantine and isolation measures were modified. Since 1 January 2021, soldiers repatriated from Sub-Saharan Africa, who tested positive for SARS-CoV-2 by RT-qPCR, are isolated in a hotel for a minimum of 10 days after the first positive test result and until symptoms have fully resolved (incl. > 3 days without fever).

SARS-CoV-2 whole-genome sequences were determined for RT-qPCR-positive nasopharyngeal samples with sufficiently high virus titers (Ct < 35), originating from soldiers repatriated from Sub-Saharan Africa, or when the presence of VOCs was suspected (Figure 1). Viral genomes were analyzed to identify possible origins and routes of spread, identify SARS-CoV-2 genetic variants, and assess the measures implemented to prevent or control COVID-19 outbreaks, overseas and at home. For the most recent infections, stemming from returning soldiers in April 2021, we find that the containment measures that have been put in place were well respected and prevented the spread into the population of lineages A.27 (a variant under monitoring) and B.1.525 (a variant of interest), as determined through phylogenetic analysis of all globally available genomes to date.

## 2. Materials and Methods

### 2.1. SARS-CoV-2 RT-qPCR

Nasopharyngeal sample collection kits (Copan Diagnostics, Murrieta, CA, USA), consisting of a flexible mini-tip flocked swab in a tube filled with 1 mL Universal Transport Medium (UTM), were used for sample collection. Quantitative RT-PCR (RT-qPCR) was performed on the samples using two methods. Until October 2020, the Novel Coronavirus (2019-nCoV) Nucleic Acid Diagnostic Kit and Sample Release Reagent (both Sansure Biotech Inc., China) were used, according to the manufacturer’s instructions, including some modifications to cater for a LightCycler 2.0 instrument (Roche Diagnostics, Diegem, Belgium) [16]. The RT-qPCR assay amplifies the ORF1ab and N genes, which encode a well-conserved replicase polyprotein and the nucleocapsid protein, respectively. Each assay also included an internal RNA control (RNAse P genes), which detects failure/inhibition of the RT-PCR. Swab samples were first inactivated at 56 °C for 30 min. From November 2020 on, SARS-CoV-2 RNA was extracted from 100 μL UTM using the Maxwell RSC 16 Instrument and the Maxwell RSC Viral Total Nucleic Acid Purification Kit (both purchased from Promega Benelux, The Netherlands). The LightCycler 480 Real-Time PCR system and the Lightmix modular SARS-CoV E-gene kit (both Roche Diagnostics Belgium) were used for reverse transcription, amplification, and detection of SARS-CoV-2, using primers and probe targeting the viral E gene. RNA extraction and RT-qPCR were performed following the manufacturers’ instructions. 

### 2.2. SARS-CoV-2 Genome Sequencing

Samples with sufficiently high virus concentrations (RT-qPCR Ct values < 35) were selected for sequencing on a MinION device using R9.4 flow cells (Oxford Nanopore Technologies, Oxford, UK), as previously described [16,17]. Briefly, using a Maxwell RSC instrument, RNA was extracted from 200 μL of UTM transport medium and eluted in 80 μL of water. Seven microliters of this suspension were used to convert SARS-CoV-2 RNA to cDNA, using random hexamers and the ProtoScript II First Strand cDNA Synthesis Kit (New England Biolabs, Hitchin, UK). A strain-specific multiplex PCR was performed in six reactions using an 800 bp SARS-CoV-2 primer scheme (Appendix A) and Q5 High-Fidelity DNA polymerase (New England Biolabs). The resulting PCR products were pooled, cleaned using AmpureXP magnetic beads (Beckman Coulter, High Wycombe, UK), and quantified using Qubit 3.0 (Thermo Fisher Scientific, Waltham, MA, USA). Samples were then barcoded using the Ultra II End Repair/dA-Tailing Module (New England Biolabs) and native barcoding kits NBD104 and NBD114 (Oxford Nanopore Technologies), and cleaned using magnetic beads, before being pooled in equimolar ratios prior to ligation of the AMII adapters with blunt/TA ligase mastermix (New England Biolabs). Sequencing libraries were loaded onto an R9.4 flow cell using a ligation sequencing kit LSK109 (Oxford Nanopore Technologies). Sequence data were collected and the Guppy algorithm v3.6 (Oxford Nanopore Technologies) was used to base call and demultiplex the sequence reads. Consensus genome sequences were produced by comparing two different bioinformatics pipelines: (1) an in-house customized pipeline, which aligns the reads with a SARS-CoV-2 reference genome (GenBank MN908947.3) using minimap2, was applied. After removal of primer sequences using a custom Python script, a majority rule consensus was produced for positions with ≥100× genome coverage, while regions with lower coverage were masked with N characters [18]; (2) the Artic network bioinformatics pipeline [19] was run, which includes both base-quality data and the Nanopolish algorithm [20] to improve the consensus sequence, for a draft genome assembly.

The protocol was slightly adjusted for the military missions ending in April 2021, in that the Artic primer pool was used along with the Artic network pipeline using medaka, which is faster than Nanopolish for variant calling at mostly equivalent performance. Finally, basecalling and de-multiplexing were done using Guppy v4.4.5.

### 2.3. Genomic Epidemiology

Whole-genome sequences were compared with the reference SARS-CoV-2 Wuhan genome (NC_045512.2) and with all SARS-CoV-2 genomes submitted to the Global Initiative on Sharing All Influenza Data (GISAID) repository (https://www.gisaid.org accessed on 6 July 2021) [21] and to the National Center for Biotechnology Information (NCBI). BLOSUM62 substitution scores were calculated to evaluate amino acid changes and their potential to alter the functionality of the proteins of interest [22]. Negative scores are predicted as “bad”, i.e., these substitutions are not often observed in nature. The obtained SARS-CoV-2 genome sequences were submitted to GISAID. The genomes correspond to returning soldiers from Niger (access IDs: EPI_ISL_487433-EPI_ISL_487436), Mali (EPI_ISL_1591101-EPI_ISL_1591104), Afghanistan (EPI_ISL_649154), the Democratic Republic of the Congo (EPI_ISL_487432), and again from Mali (EPI_ISL_2404564-EPI_ISL_2404570, and EPI_ISL_2425330-EPI_ISL_2425331). Importantly, travel history for each infected individual was added to the corresponding GISAID entry in the “Additional location information” field so that this information can be exploited in various data analyses [23].

In order to perform a nucleotide and amino acid comparison between all generated SARS-CoV-2 genomes they were placed in a global SARS-CoV-2 phylogeny using UShER on 28 May 2021. The resulting auspice JSON was processed in baltic v.0.1.5 (https://github.com/evogytis/baltic accessed on 6 July 2021) to rename clades from Nextstrain nomenclature to pangolin nomenclature (20J/501Y.V3 as P.1, 20H/501Y.V2 as B.1.351, 20I/501Y.V1 as B.1.1.7, 20E (EU1) as B.1.177, 20A.EU2 as B.1.160, 20D as C.16, 20G as B.1.2, and 20F as D.2) and collapsed. Additional clades not yet recognized in Nextstrain nomenclature—B.1.617.2, A.23.1, and B.1.526—were identified and collapsed as well. All other tips in the phylogeny were trimmed out unless their pangolin designation matched the pangolin designation of any study sequence. 

These genomes were first analyzed in Nextstrain [24], an open-source project that provides a continuously updated phylogenetic analysis of the evolution and relationships of publicly available SARS-CoV-2 genomes. Nextstrain consists of an automated pipeline that includes subsampling, alignment, maximum-likelihood phylogenetic inference, temporal dating of ancestral nodes, and ancestral geographic location reconstruction. The Nextstrain phylogeny is typically rooted by using the 2019 Wuhan samples as outgroup sequences [24]. Given that the Nextstrain application generates a frequency plot over time of all the SARS-CoV-2 lineages that make up its data set, we have created custom frequency plots for the purpose of this study. To this end, we retrieved all African SARS-CoV-2 genomes and lineage annotations from GISAID on 21 May 2021, and calculated the proportion of sequences collected in a given day for each lineage. The resulting time series were smoothed using a second-order Savitzky–Golay filter [25] on 61-time windows, as implemented in the SciPy Python library [26].

SARS-CoV-2 lineages were designated using the dynamic nomenclature system proposed by Rambaut et al. [27] and are referred to as “pango lineages”. We here reiterate the pangolin nomenclature scheme for clarity. Major lineage labels begin with a letter. At the root of the SARS-CoV-2 phylogenetic tree, based on sequences submitted to the GISAID database, are two lineages that were denoted as lineages A and B. Further SARS-CoV-2 lineages, descending from either lineage A or B, were assigned a numerical value (e.g., lineage A.1 or B.2). Lineages are designated according to the following rules: (1) each descendant lineage should show phylogenetic evidence of emergence from an ancestral lineage into another geographically distinct (e.g., a new region, province, or country) population, implying substantial onward transmission; (2) these newly identified lineages can themselves be ancestors for lineages that then emerge in other geographical areas or at later times (e.g., A.1.1); (3) this procedure can proceed for a maximum of three sublevels (e.g., A.1.1.1), after which new descendant lineages are given a letter in sequence from C (e.g., A.1.1.1.1 would become C.1). 

SARS-CoV-2 variants were assigned to “Nextstrain clades” using the Nextclade tool [28]. Major clades are named after the year in which they are assumed to have originated, and a letter (e.g., 19A, 19B, 20A). Within these major clades, “emerging clades” are labeled with their parent clade and the nucleotide mutation(s) that define(s) them (e.g., 19A/28688C). When such a “subclade” reaches a frequency of 20% globally, it is renamed to a “major clade” (e.g., 19A/28688C to 20D). The first two clades are 19A and 19B. Both clades were common in Asia during the first months of the outbreak. The next major clade, which dominated major European outbreaks in the early 2020s, was named 20A. Two more clades appeared later on in 2020: clade 20B, another “European clade”, and clade 20C, a largely “North American clade”. For a while, no new clades exceeded the 20% global frequency. Instead, “regional clades” achieved significant frequencies in different regions of the world. An example was 20A.EU1, which expanded in Europe. In an updated strategy, major (year letter) clades are called when a clade reaches > 20% global frequency for 2 or more months, or a clade reaches > 30% regional frequency for 2 or more months. At the end of 2020 and beginning of 2021, three VOCs emerged in the U.K. (20I/501Y.V1), South Africa (20H/501Y.V2), and Brazil (20J/501Y.V3) [29]. Currently, Nextclade defines 12 major clades.

### 2.4. Phylogenetic Inference

We first created an alignment of all genomes obtained from infected military personnel, complemented with the reference SARS-CoV-2 Wuhan genome (NC_045512.2), in order to create a custom visualization of the nucleotide and amino acid differences between these genomes. We subsequently performed detailed phylogenetic analysis on the data of the Belgian soldiers who returned from Mali in April 2021. Given that the genomes identified from some of these soldiers were classified as A.27 and B.1.525, we constructed two separate data sets of all globally available genomes on GISAID on the 1st and 9th of May 2021, respectively. All data sets were aligned using MAFFT v7.475 [30], followed by visual inspection and corrections, and 5′ and 3′ untranslated regions of the genome that were susceptible to sequencing and assembly error trimmed. We subsequently performed an iterative TempEst [31] analysis to remove genomes that had either too few or too many private mutations, building an updated unrooted phylogeny using IQ-TREE v2.1.2 [32] at each step. The final step of this analysis consisted in building a consensus phylogeny using the UltraFast bootstrap option in IQ-TREE, which was then used as input for TreeTime [33] to obtain a time-calibrated phylogeny. We performed the TreeTime analysis using an autocorrelated molecular clock model with a standard deviation of 0.0004 and a skyline coalescent model [33]. The resulting phylogeny was visualized using ggtree [34].

### 2.5. Ethics Statement

Clinical samples were routinely processed in view of diagnosis. Results were analyzed in the context of retrospective epidemiological analysis and were not deemed prospective research.

## 3. Results and Discussion

During the one-year study period (May 2020–April 2021), Belgian soldiers involved in missions and operations were tested for SARS-CoV-2 infection, pre- and post-mission, using RT-qPCR (*n* = 7683). Twelve soldiers who were isolated for 14 days prior to deployment to missions or operations tested positive for SARS-CoV-2 two days before their departure. This reinforced the belief in the usefulness of the implemented pre-mission quarantine procedure. Upon return to Belgium, 21 soldiers tested positive for SARS-CoV-2. These soldiers had returned from missions in Niger (Maradi, 19 May 2020, *n* = 4), the Democratic Republic of the Congo (DRC) (Kinshasa, 2 June 2020, *n* = 1), Afghanistan (Kabul, 30 June 2020, *n* = 1), and Mali (Bamako, 8 November 2020, *n* = 6; Gao, 1 April 2021, *n* = 7; and Bamako, 12 April 2021, *n* = 2).

Most SARS-CoV-2 infected soldiers returning from overseas missions were relatively young and fit men and women, and were either asymptomatic or experienced mild symptoms. Of note, asymptomatic SARS-CoV-2-positive soldiers are often persistent SARS-CoV-2 carriers and spreaders [14]. However, a 32-year-old soldier, who returned from Afghanistan, suffered from severe fever, watery diarrhea, fatigue, muscle pain, and general malaise, and was admitted to a hospital in Kabul for oxygen treatment prior to his repatriation to Belgium. In Belgium, he did not require further hospitalization, but experienced fatigue and anosmia for several weeks. A 55-year-old military expat, who was infected in the DRC, was quarantined at the Queen Astrid Military Hospital in Brussels for three weeks, for social reasons.

To analyze possible patterns of spread, especially among the small clusters of SARS-CoV-2-positive soldiers returning from Niger [16] and Mali, and to analyze the possible introduction of new SARS-CoV-2 variants in Belgium, we further determined the whole-genome sequences of the SARS-CoV-2 isolates. Nineteen complete genome sequences were obtained from the 21 RT-qPCR-positive samples. The viral load in the nasopharyngeal swabs from two soldiers returning from Mali (Bamako, 8 November 2020) was too low (Ct > 37) for reliable whole-genome sequencing.

We identified nine distinct genotypes and arbitrarily named them “Niger 1–3”, “DRC 1”, “Afghanistan 1”, and “Mali 1–4” for the purpose of this study. Importantly, Figure 2 shows the key mutations among the infected military personnel, which are discussed in the sections below, organized per military mission abroad.

As the SARS-CoV-2 lineages identified are clearly circulating on the African continent, we provide an overview of the predominant lineages in Africa in Figure 3. While no soldiers returned with a B.1.1.7 nor B.1.351 infection, we also highlight those numbers for context. Figure 3 shows that none of the lineages that infected Belgian military personnel are currently becoming dominant, although the first trimester of 2021 showed a clear increasing trend of B.1.525 and A.27 infections, which were likely pushed down because of the rise in B.1.351 and B.1.1.7 infections, which jointly make up the majority of infections on the African continent since the start of 2021.

Figure 4 shows the global SARS-CoV-2 phylogeny, obtained by augmenting the global Nextstrain build (https://nextstrain.org/ncov/global accessed on 6 July 2021) with the genomes obtained in this study. Returning soldiers are shown to be infected with a wide range of SARS-CoV-2 lineages, strongly linked to the timing and location of their mission abroad. We discuss each of these missions and lineages in more detail below.

### 3.1. Niger 1–3 (Maradi, 19 May 2020)

With two identical genomes, one that differed by a single-nucleotide polymorphism (SNP), and another by two SNPs, the restricted genomic variations observed in the four soldiers, who had been infected during an outbreak in a military training camp in Niger, were consistent with those observed in “institutional outbreaks” [15,16]. The three Niger genotypes (Niger 1–3) belong to pango lineage A and Nextstrain clade 19B (Figure 2 and Figure 3), which was first observed in travelers from Wuhan in the early days of the outbreak [4] and became common in North America and Europe [5,35,36], but has now declined worldwide (Figure 3). This clade was prevalent in West Africa when the Belgian soldiers were present in Niger (February–May 2020) [16]. The three Niger genomes carry the two distinctive mutations of clade 19B (C8782T and T28144C) and the C18959T mutation, which was unique to the Niger genomes and results in an A1831V amino acid change in ORF1b (Figure 1). All Niger genomes harbored the non-synonymous mutation G254R in ORF3a, which encodes the SARS-CoV-2 protein 3a that was shown to form homotetrameric ion channels (viroporins) localized at endosomes and lysosomes, which may allow virus delivery, and was linked to infectivity and virulence [37,38]. The mutation that was expected to have the most significant impact (with a BLOSUM score of −3), G20477T, resulting in a G2337V amino acid substitution, was found in the Niger 2 genome, but has only been observed once so far, in March 2020 [16]. It is thus considered unlikely that this mutation affects the infectivity of the virus. 

### 3.2. DRC 1 (Kinshasa, 2 June 2020)

The DRC 1 genotype belongs to pango lineage B.1.214 and Nextstrain clade 20A (Figure 2 and Figure 3) and contains the signature mutations P314L in ORF1b and D614G in the spike (S) protein. The P314L amino acid substitution is located in an interphase region of nsp12, the catalytic domain of the SARS-CoV-2 polymerase [39,40]. It was suggested that it could confer a structural alteration and an adverse effect on proofreading during the genomic replication of the virus. The P314L substitution is located in a pocket that has been predicted as a possible druggable site, and further research is needed to evaluate if the mutation could affect these properties. The D614G mutation is a major circulating spike mutation of SARS-CoV-2. It has been suggested that the D614G amino acid substitution is associated with potentially higher viral loads, but not with disease severity [9,41,42]. Although the D614G mutation occurs in the spike protein, it does not seem to significantly affect vaccine efficacy or influence antibody-induced immunity, as the receptor-binding domain of the virus is not affected by this locus. A study conducted on 18,514 sequences suggested that vaccines based on the Wuhan reference strain were likely to be effective against the then (September 2020)-circulating variants [43]. 

### 3.3. Afghanistan 1 (Kabul, 30 June 2020)

The Afghanistan 1 genotype was assigned to pango lineage B.1.1.254 and Nextstrain clade 20B (Figure 2 and Figure 3), and harbors the characteristic mutations G28881A, G28882A, and G28883C that confer R203K and G204R amino acid changes in the N gene, and a G50N amino acid substitution in ORF14, depending on the translation frame. This GGG to AAC mutation at genome positions 28881-3 overlaps the first three bases of the 5′ end of the Chinese CDC N gene forward primer (Figure 2) [44,45]. This mismatch impairs N-gene RT-qPCR, but since the Chinese CDC assay also includes an ORF1ab RT-qPCR, this lineage will still be detected by routine diagnostic PCR. Nevertheless, it seems prudent not to use this RT-qPCR primer. The Afghanistan 1 genotype also harbors an L3606F amino acid substitution in the transmembrane domain of nsp6 and the previously discussed D614G substitution, two of three changes found across all geographic and climatic zones, indicating that they are particularly important for universal infectivity and vaccine development [46]. The Afghanistan 1 genotype also harbored an I35T amino acid change in the previously discussed (Niger 1–3) ion-channel-forming protein 3a [37,38].

### 3.4. Mali 1 (Bamako, 8 November 2020)

The four identical genomes extracted from four soldiers returning from Mali suggest an infection cluster with a very short transmission chain, comparable to those observed within households [47], but also a potentially faster transmission ability of this genotype. The Mali 1 genotype was shown to belong to pango lineage B.1 and Nextstrain clade 20A (Figure 2 and Figure 3). This lineage carries the previously discussed P314L amino acid substitution in ORF1b and a Q57H mutation (Figure 2), which is ubiquitous and is consistently the most frequent ORF3a substitution described in the literature [37,38]. Although this site is lining the transmembrane tunnel, the results of site-directed mutagenesis experiments demonstrated no alteration of the “viroporin” channel properties [37]. The Mali 1 genotype contains the above-discussed and now widespread D614G spike mutation, but also spike mutation Q675H, which is readily observed in VOCs and is located in the proximity of the polybasic cleavage site (residues 682–685) at the S1/S2 boundary influencing the binding of the receptor-binding domain (RBD, a subunit of the S glycoprotein) and the angiotensin-converting enzyme 2 (ACE2) receptor [48]. This mutation may hinder efficiency of existing vaccines and may subsequently expand in response to the increasing after-infection or vaccine-induced seroprevalence [48]. When the soldiers “carrying” this lineage arrived in Belgium, on 8 November 2020, the S:Q675H mutation had only been reported a handful of times in Belgium. Indeed, a quick increase of this mutation was observed during October 2020, associated with an increase in infections of lineage B.1.221. While this mutation was not reported in Belgium during December 2020, the subsequent months of January and February in 2021 saw further accumulation of infections with the S:Q675H mutation stemming from infections with B.1.177 and to a lesser extent B.1.36. During these two time periods, a small number of infections with lineage B.1 was also observed, however. On 10 June 2021, this mutation had been observed a total of 111 times in Belgium. Finally, a C/T mismatch was observed between the variant’s genome sequence and the first position (28833) at the 5′ end of RT-qPCR reverse primer Charité_N_R, targeting positions 28814–28833 of the nucleocapsid (N) gene (Figure 2). This mismatch was shown to be particularly prevalent in Western Africa [49] and it was not reported to what extent it impairs SARS-CoV-2 detection.

### 3.5. Mali 2 and 3 (Gao, 1 April 2021)

Six of the seven SARS-CoV-2 genomes retrieved from soldiers repatriated from Gao were identical (Mali 2, Figure 2), again indicative of a very short transmission chain [47]. One genome (Mali 3) differed from Mali 2 by a single SNP, which resulted in an L13F amino acid replacement in the membrane (M) protein, which was shown to suppress type I and III interferon (IFN)-mediated antiviral immunity [50]. The Mali 2 and 3 genotypes were assigned to pango lineage A.27 and Nextstrain clade 19B (Figure 2 and Figure 3). Of note, only 7 sequences in the CoV-GLUE Amino acid variation database contained the M:L13F mutation [51], and none of them belonged to lineage A.27. The A.27 genotype is characterized by the previously discussed D614G mutation in the spike protein, but also by four other missense mutations in this protein, which were potentially linked to increased transmissibility and/or decreased immune neutralization: S:L18F in P.1 and in over 35% of B.1.351; S:L452R in B.1.617.2; S:N501Y in B.1.1.7, B.1.351, and P.1; and S:H655Y in P.1 (Figure 2) [52,53,54]. Recent studies indicate that the S:L452R mutation is of significant adaptive value to SARS-CoV-2 [53,54]. It was also suggested that the strong and recent positive selection for this mutation reflects viral adaptation to the containment measures or increasing population immunity [53]. Leucine-452 is located in the RBD, which binds to the ACE2 receptor, and its replacement with arginine is predicted to result in a much stronger binding to the receptor and an escape from neutralizing antibodies [53]. If this proves to be true, L452R variants might exhibit an increased infectivity, calling for a heightened vigilance.

This A.27 lineage, which emerged in December 2020, was classified by the European Centre for Disease Prevention and Control (ECDC) as a “variant under monitoring” [55]. It is currently unclear where it originated, but today it is mostly detected in France [27], and recently, it was shown to be on the rise in Côte d’Ivoire [56] and spreading quickly in Germany [57]. The genomes collected in this study point to A.27 currently circulating in Mali, with seven Belgian soldiers having returned from Mali in April 2021 and, upon arrival, being found to be infected with A.27 (Figure 5). Their genomes cluster within a nearly entirely French cluster, which can be attributed to a large collection of French A.27 genomes and no A.27 genomes from Mali being available. Importantly, no novel Belgian A.27 genomes have been found since, through baseline surveillance (unbiased sampling with a coverage of approximately 5% since the start of 2021), that could be linked to the infected soldiers returning from Mali. This provides evidence that the post-mission containment strategy deployed by the Belgian military is safe, being adhered to, and, as a result, prevents infections from entering the general population.

To obtain a better understanding of the circulation of lineage A.27 in West-African countries, we analyzed the travel history for the non-military Belgian A.27 genomes that were obtained in the context of unbiased baseline surveillance. We found three such travel-related cases, which can be seen in Figure 5, near the top of the phylogeny. Two samples originated from returning travelers from Burkina Faso, while another person was most likely infected in Belgium by his daughter who had traveled to Burkina Faso and tested positive upon her return to Belgium. All three genomes cluster closely with one of only two available A.27 genomes from Burkina Faso and an A.27 genome from Togo, providing further evidence as to the circulation of lineage A.27 in the three neighboring countries of Mali: Burkina Faso, Togo, and Ivory Coast. It is thus fair to assume that A.27 may also be circulating locally in Ghana, even though Ghana has not reported any A.27 genomes so far during their sequencing efforts.

### 3.6. Mali 4 (Bamako, 12 April 2021)

The Mali 4 genotype, retrieved from two military service men, was assigned to pango lineage B.1.525 (also known as VUI-21FEB-03, VUI-202102/03, or UK1188) (Figure 2 and Figure 3) and Nextstrain clade 20A [58]. It carries the E484K and Q677H mutations, and the ΔH69/ΔV70 deletion in the spike protein which is also found in B.1.1.7, the variant that was first identified in the UK. The deletion at positions 69–70 of the spike protein was shown to affect the performance of some diagnostic PCR assays that use the S gene as a target [5]. Lineage B.1.525 distinguishes itself from all other lineages by having both the S:E484K mutation and a new S:F888L mutation in the S2 domain of the spike protein. The ECDC classified lineage B.1.525 as a “variant of interest” (VOI) [55], exhibiting reduced antibody neutralization [59,60]. The first cases were detected in December 2020 in the UK, North America, Angola, and Nigeria [29,55] and, since then, more than one third of SARS-CoV-2 sequences have been assigned to lineage B.1.525 in Nigeria [29]. On the 10th of June, 2021, this VOI had been detected in 49 countries, including Belgium (*n* = 55).

Figure 6 shows that Belgian genomes can be found in nearly all clades of the B.1.525 phylogeny. The two identical genomes corresponding to the returning Belgian military men are indicated by the red asterisk and reside in a large multifurcating clade with little structure. These two Belgian sequences cluster as a sister clade to a collection of mostly Swiss and English genomes, and one German and Nigerian genome. Again, no B.1.525 genomes from Mali were available on GISAID. Importantly, a total of 233 B.1.525 genomes are available from the continent of Africa, including from neighboring countries of Mali such as the Ivory Coast, Togo, Guinea, and Ghana. Combined with the data presented here, and similar to A.27, this provides evidence of B.1.525 actively circulating in Mali.

Given the difficulties in visualizing such a large collection of genomes for lineage B.1.525, we constructed a dedicated Nextstrain build to enable a thorough exploration of our results: https://nextstrain.org/community/GuyBaele/sars-cov-2-belgium/B.1.525 (accessed on 8 June 2021). GISAID entries corresponding to genomes with imprecise sampling dates were discarded to generate a sufficiently responsive Nextstrain instance and to limit the file sizes (to <100 Mb) so that the visualization could still be made publicly available.

## 4. Conclusions

Our study showed that during the period from May 2020 to April 2021, 21 Belgian soldiers returned to Belgium from overseas missions in Niger, the DRC, Afghanistan and Mali harboring nine distinct SARS-CoV-2 genotypes, belonging to three major Nextstrain clades (19B, 20A and 20B) (Figure 2 and Figure 3). These imported variants exhibited dozens of missense mutations (Figure 2), some of which are predicted to elicit increased infectivity and/or decreased drug and vaccine efficacy. Especially, VOI B.1.525 and “variant under monitoring” A.27 call for a heightened vigilance. Three variants harbored mutations causing mismatches with established RT-qPCR primers targeting the N- and S-genes, which could hamper the sensitivity of diagnostic tests. The identified SARS-CoV-2 lineages were already present in Belgium before the return of the soldiers, and there are no indications that they transmitted any variants to their military or civilian environments. The implemented COVID-19 risk mitigation policies, and more specifically pre-departure and early post-mission isolation and testing, appeared to have prevented both the export and import and spread of SARS-CoV-2 variants, while safeguarding most of the readiness of the Belgian Armed Forces and their ability to conduct missions and operations. Additionally, the low number of infections among returning military personnel is also a testament to the imposed measures during those missions. Nevertheless, we must continue to guard against the spread, though travel (incl. military missions) of certain variants, which may later prove to be more infective, resistant to drugs or vaccination, and/or evade diagnosis.

Our study shows the importance of sequencing samples from returning travelers and keeping track of metadata associated with these travelers so that they can be made available on GISAID. Novel phylogeographic analysis methods [23] are able to exploit such individual travel histories to mitigate the effects of undersampling—or not sampling at all—certain locations when reconstructing the spread of SARS-CoV-2 lineages. Associated metadata can hence contribute to the accuracy of phylogeographic inference in the absence of genomic data, as these data can be used to mitigate sampling bias (to a certain extent). Such methods are seeing increased application for various lineages, such as A.23.1 [61] and B.1.380 [61], B.1.620 [62] and A [63].

Finally, this study underscores the importance of a timely monitoring of genetic changes and investigating the relationship between variants and transmissibility, disease severity, and the performance of diagnostic methods, antiviral drugs and vaccines, in a pandemic situation, and as a foundation for future worldwide genomic surveillance of infectious diseases.

## Figures and Tables

**Figure 1 viruses-13-01359-f001:**
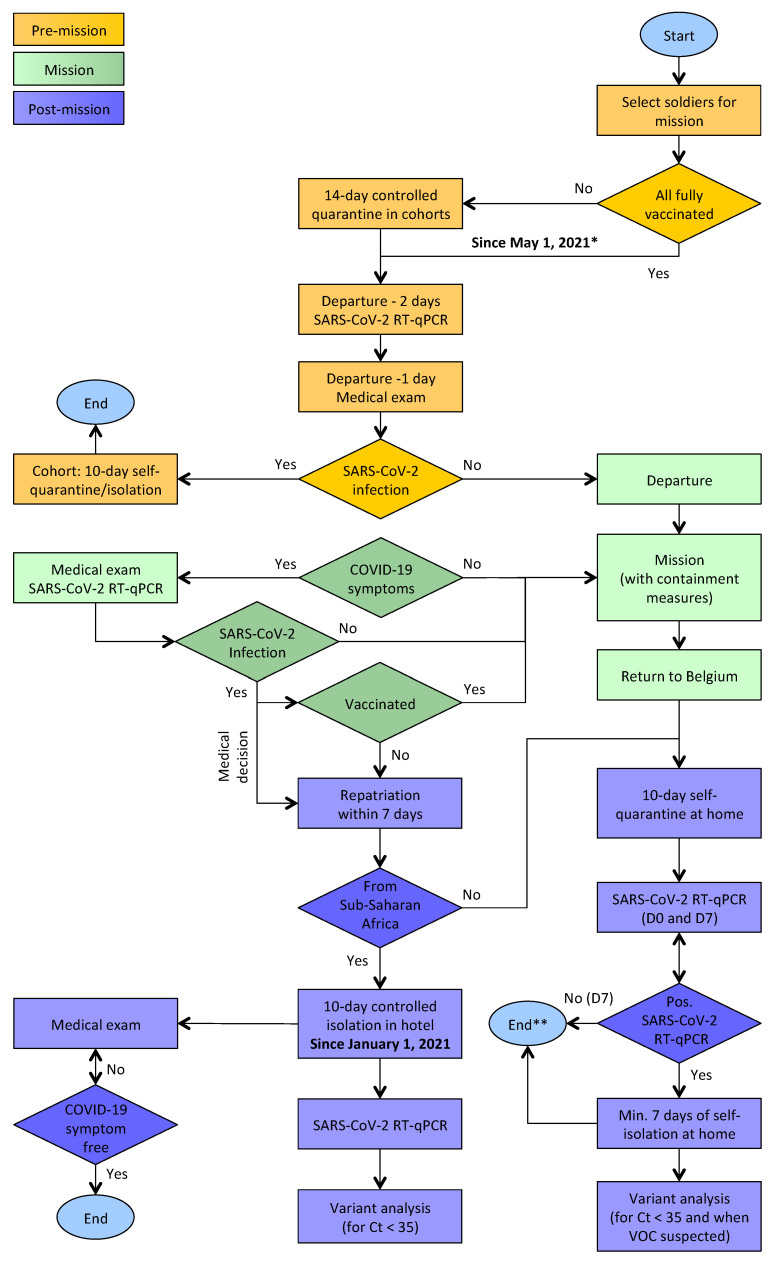
Flowchart describing Belgian Armed Forces deployment to international military operations, with an emphasis on coronavirus risk mitigation measures. Both 1 January and 1 May 2021 marked changes in the implementation of these measures, the former in response to the emergence of variants of concern (VOCs), the latter in response to the ongoing vaccination program in Belgium. * Unless the destination country or the mission’s lead nation request otherwise. ** Quantitative RT-PCR (RT-qPCR) test results are known within 1–3 days after sampling, but everyone has to self-quarantine/isolate for at least 10 days.

**Figure 2 viruses-13-01359-f002:**
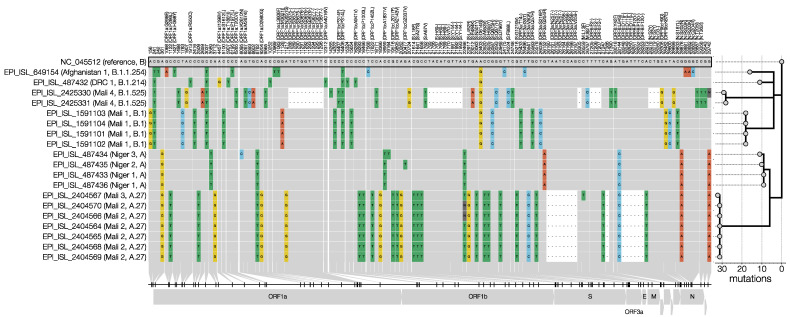
Nucleotide and amino acid comparison of imported genomes to the reference severe acute respiratory syndrome coronavirus 2 (SARS-CoV-2) Wuhan genome (NC_045512.2). Only single-nucleotide polymorphisms (SNPs) that differentiate the genomes from the reference are shown in the condensed SNP alignment. Sites identical to the reference are shown in grey, changes from the reference are indicated and colored by nucleotide (green for thymidine, red for adenosine, blue for cytosine, yellow for guanine, and black for gaps). If a mutation results in an amino acid change, the column label indicates the gene, reference amino acid, amino acid site, and amino acid change in brackets. The unrooted phylogeny (branch lengths number of mutations) on the right shows the relationships between depicted genomes and was rooted on the reference sequence.

**Figure 3 viruses-13-01359-f003:**
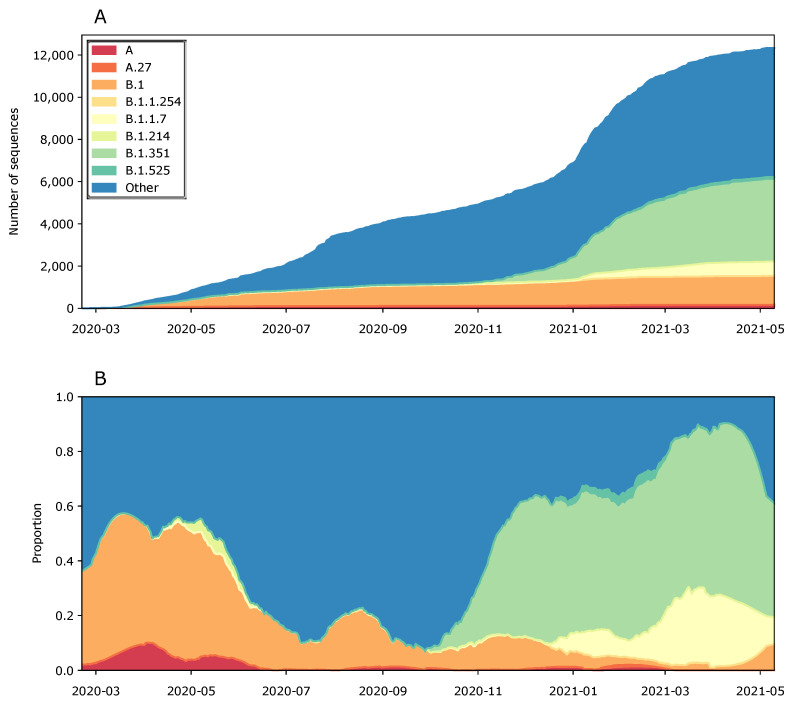
(**A**) Cumulative number over time of African severe acute respiratory syndrome coronavirus 2 (SARS-CoV-2) sequences by lineage. (**B**) Frequencies over time of the key lineages identified in this study, along with the B.1.1.7 and B.1.351 variants of concern. Regarding the lineages in this study, we note that their impact in the pandemic is currently rather limited, due to the huge number of infections attributed to the B.1.351 lineage.

**Figure 4 viruses-13-01359-f004:**
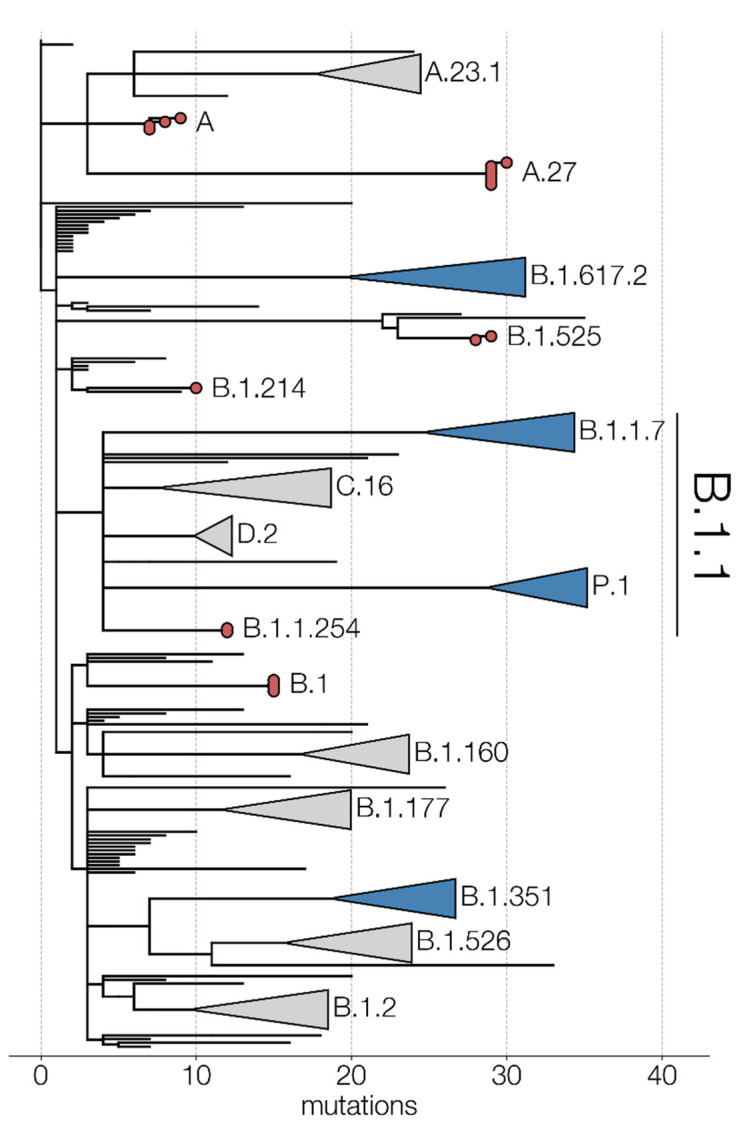
Severe acute respiratory syndrome coronavirus 2 (SARS-CoV-2) genomes from Belgian military personnel (in red) on a global Nextstrain phylogenetic analysis (accessed on the 27th of May, 2021; divergence view) focusing on the major lineages of SARS-CoV-2 genomes submitted to the Global Initiative on Sharing All Influenza Data (GISAID) repository. Blue triangles represent collapsed SARS-CoV-2 variant of concern (VOC) clades and grey triangles other major SARS-CoV-2 clades in the phylogeny.

**Figure 5 viruses-13-01359-f005:**
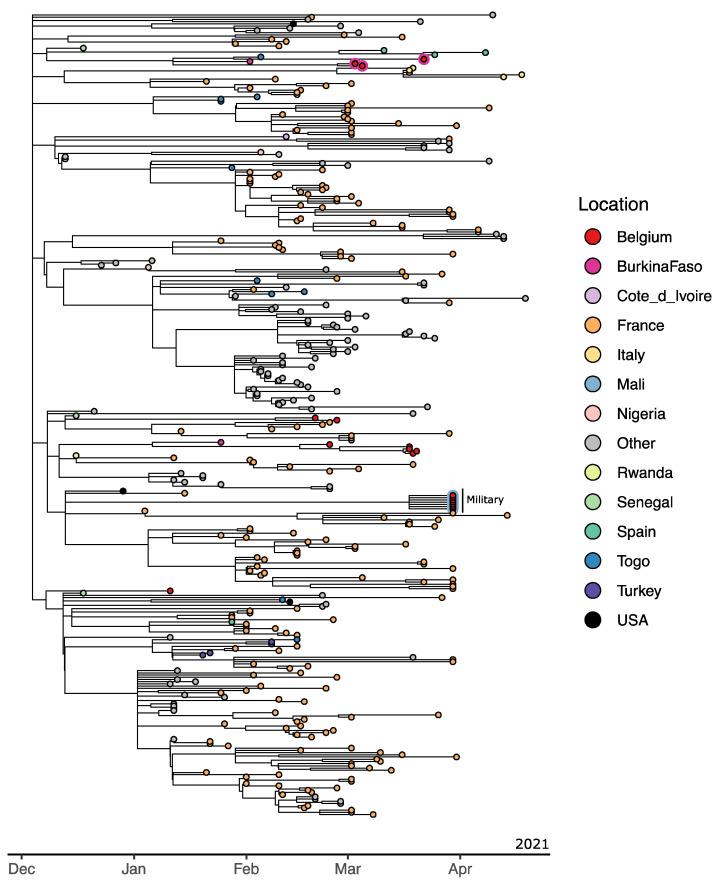
Time-calibrated A.27 phylogeny highlighting Belgian genomes with known travel history. Tips are colored according to country of sampling, with additional circumferences indicating country of infection. The seven A.27 genomes from Belgian soldiers returning from Mali (red circle with light blue outline) are found to cluster within a group of French A.27 genomes. Three non-military Belgian genomes were found to have associated travel history to Burkina Faso (red circles with purple outline).

**Figure 6 viruses-13-01359-f006:**
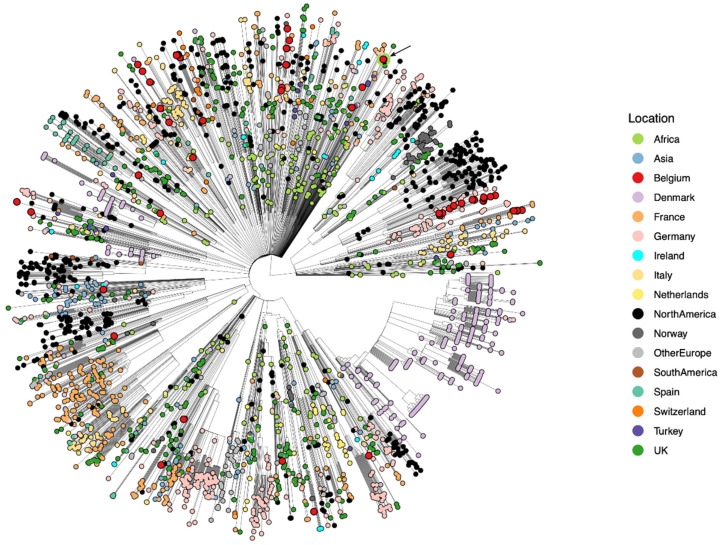
Large-scale time-calibrated phylogeny of all globally available B.1.525 genomes, visualized using a circular representation due to the large number of genomes. Belgian genomes are represented with larger (red) tips, with the two Belgian military service men indicated by the arrow (identical genomes on the same collection date). Tips are colored according to country of sampling, with additional circumferences indicating country of infection. Belgian genomes can be found across the entire phylogeny, but no infections found through baseline surveillance in Belgium stem from the infected soldiers returning to Belgium.

## Data Availability

The SARS-CoV-2 genome sequences generated during this study were submitted to the Global Initiative on Sharing All Influenza Data (GISAID) repository (https://www.gisaid.org accessed on 6 July 2021). Their access IDs are: EPI_ISL_487433-EPI_ISL_487436, EPI_ISL_159110-EPI_ISL_1591104, EPI_ISL_649154, EPI_ISL_487432, EPI_ISL_2404564-EPI_ISL_2404570, and EPI_ISL_2425330-EPI_ISL_2425331.

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
