# Peer review of "Variant Analysis of SARS-CoV-2 Genomes from Belgian Military Personnel Engaged in Overseas Missions and Operations"

_viruses, 2021, doi:10.3390/v13071359_

Round 1
Reviewer 1 Report
Proposed paper is a really interesting and really well written one. I have only two minor point to be addressed before it could be accepted for pubblication:
- There are very few number of cases and this is surely a strenght of the protocol of the belgian military agency. Although it has been already discussed please reinforce this concept in the discussion.
- A very recently published review on COVID have been missed and could be cited in the introduction (i.e. High Blood Press Cardiovasc Prev. 2021 Jun 26:1–7. )
Author Response
Dear reviewer,
We appreciate very much your time in reviewing our manuscript.
We reinforced the Belgian military concept in the manuscript and added the proposed reference.
Many thanks.
Kind regards,
Jean-Paul Pirnay
Reviewer 2 Report
The authors assessed 7,683 samples obtained from Belgian soldiers involved 305 in missions and operations during May 2020 - April 2021. Twelve soldiers tested positive for SARS-CoV-2 two days before their departure and upon return to Belgium, 21 soldiers tested positive.
This manuscript is well-written, the figures are of high quality, and the authors have clearly worked hard to produce a comprehensive dataset and detailed description of their methods. I would recommend it for acceptance after one minor point below.
Line 140: Do you mean 2021?
Author Response
Dear reviewer,
We would like to thank you for the time you spent reviewing our manuscript.
Many thanks for observing the wrong year in line 140. We corrected this in the manuscript.
Kind regards,
Jean-Paul